# Immuno-Neutralization of Follistatin Bioactivity Enhances the Developmental Potential of Ovarian Pre-hierarchical Follicles in Yangzhou Geese

**DOI:** 10.3390/ani12172275

**Published:** 2022-09-02

**Authors:** Rong Chen, Pengxia Yang, Zichun Dai, Jie Liu, Huanxi Zhu, Mingming Lei, Zhendan Shi

**Affiliations:** 1Institute of Animal Science, Jiangsu Academy of Agricultural Sciences, Nanjing 210014, China; 2Jiangsu Key Laboratory for Food Quality and Safety-State Key Laboratory Cultivation Base, Ministry of Science and Technology, Nanjing 210014, China; 3Key Laboratory for Crop and Animal Integrated Farming, Ministry of Agriculture and Rural Affairs, Nanjing 210014, China; 4College of Animal Science and Technology, Nanjing Agricultural University, Nanjing 210095, China

**Keywords:** geese, follistatin, immunization, ovarian follicles, gene expression

## Abstract

**Simple Summary:**

Follistatin involves in the regulation of ovarian follicular development in mammals; however, the role of follistatin in goose ovarian follicular development has not been investigated. In this study, following immuno-neutralization of follistatin bioactivity in geese, the number of ovarian pre-ovulatory follicles significantly increased, and mRNA levels of genes involved in ovarian steroidogenesis and yolk deposition were upregulated in the granulosa layer of pre-hierarchical follicles. These results suggest that follistatin plays a limiting role in the development of ovarian pre-hierarchical follicles into pre-ovulatory follicles. These results also expand our understanding of the mechanism of follistatin on ovarian follicular development in geese.

**Abstract:**

In order to explore the role of follistatin (FST) in ovarian follicular development and egg production in Yangzhou geese, sixty-four egg laying geese of the same genetic origin were selected and divided into two groups with equal numbers. One group was immunized against the recombinant goose FST protein by intramuscular injection, whereas the control group received bovine serum albumin (BSA) injection. Immunization against FST significantly increased the number of pre-ovulatory follicles. Furthermore, immunization against FST upregulated *Lhr*, *Star*, *Vldlr*, *Smad3*, and *Smad4* mRNA levels in the granulosa layer of pre-hierarchical follicles. The results suggest that FST plays a limiting role in the development of ovarian pre-hierarchical follicles into pre-ovulatory follicles by decreasing follicular sensitivity to activin in geese. The mechanism may be achieved by regulating the SMAD3 signaling pathway, which affects progesterone synthesis and yolk deposition in pre-hierarchical follicles.

## 1. Introduction

Follistatin (FST), a glycosylated single-chain protein, was first identified in the follicular fluid of cattle and pigs by its ability to suppress FSH secretion in pituitary cell cultures [1,2]. Subsequently, FST has been discovered to be an activin-binding protein [3], and the activin–FST complex consists of one activin dimer and two FST molecules [4,5]. Therefore, the biological actions of FST are thought to be mediated mainly by neutralizing the bioactivity of activin. An increasing body of evidence shows that, in addition to activins, FST interacts with other TGF-β family members in the ovary, presumably through a similar binding mechanism [6].

In mammalian species, *Fst* mRNA abundance increases as follicles develop from small antral to pre-ovulatory follicles and declines during the atretic process [7,8,9,10]. However, primordial follicles do not express *Fst* mRNA [11,12]. Ovarian granulosa cells are the main site responsible for FST production and secretion. *Fst*-null mice survive through birth but die within hours of delivery [13]. Conversely, transgenic mice overexpressing FST survive to adulthood; however, defects were found in the gonad [14]. In male transgenic mice, the testes are generally smaller and show variable degrees of Leydig cell hyperplasia and seminiferous tubule degeneration. In female transgenic mice, uteri are thin and ovaries are small because folliculogenesis is blocked at the early primary or secondary (antral) follicle stages. Moreover, FST immunization in Hereford-cross heifers resulted in an increase in the number of small follicles at the time of wave emergence [15]. These results suggest that FST plays a limiting regulatory role in the development of ovarian follicles.

In birds, *Fst* mRNA abundance was found to be greater in smaller pre-hierarchical follicles than in larger pre-ovulatory follicles, and the greatest expression occurred in small yellow follicles [16,17], which is incongruous with that in mammals. Follicles in avian ovaries are arranged in a hierarchical order and consist of pre-hierarchical large white follicles (LWFs), small yellow follicles (SYFs), and hierarchical large yellow follicles (LYFs, also called pre-ovulatory follicles). The highest abundance of *Fst* mRNA was observed in SYFs, suggesting that FST may play a critical role in early follicular development, especially during follicle selection by regulating activin availability. In addition, FST may also play a significant role in the regulation of egg production, since egg production predominantly depends on the number of pre-hierarchical follicles selected into the hierarchical developmental stage [18]. Domestic geese have lower egg laying performance than chickens and ducks and possess a unique ovarian follicle development mode [19]. In the present study, we investigated the effects of immunization against FST on the recruitment of pre-hierarchical follicles, growth of pre-ovulatory follicles, and egg production in Yangzhou geese, aiming to explore a method to improve egg laying performance.

## 2. Materials and Methods

### 2.1. Ethics

Animal experiments were approved by the Jiangsu Academy of Agricultural Sciences Experimental Animal Ethics Committee and carried out according to the Regulations for the Administration of Affairs Concerning Experimental Animals (Decree No. 63 of the Jiangsu Academy of Agricultural Sciences, 8 July 2014). 

### 2.2. Immunogen

The coding sequence of a 148-amino-acid peptide of goose FST (residues 93–240, NCBI Reference Sequence: XP_013036067.1) was cloned into the pRSET-A vector between BamHI and KpnI sites (Invitrogen, Carlsbad, CA, USA). The recombinant protein was expressed and purified as described previously [18]. SDS-PAGE and Western blot were performed to analyze the protein purity. The immunogen (1 mg/mL) was prepared by emulsifying recombinant proteins with mineral oil (Solarbio, Beijing, China). Bovine serum albumin (BSA) was emulsified in mineral oil as well and used as control. Both recombinant FST protein and BSA were dissolved in physiological saline.

### 2.3. Animal Experiments

The experiments were conducted from September to November at Anhui Tianzhijiao Goose Industry Co., Ltd (117°99′ E, 32°07′ N), Chuzhou, Anhui Province, China. A flock of approximately 1-year-old Yangzhou geese in the late stage of out-of-season laying were of the same genetic origin and were exposed to a 12 h daily photoperiod (12 L: 12 D) as described previously [20]. Of these, 64 individuals were randomly divided into two groups with equal numbers. All individuals were kept in the same goose barn. For each group, 32 individuals were kept together in a pen (6 m × 4 m). After 1 week of adaptation, the experimental group was intramuscularly administered FST antigen (1 mg) on day 1, whereas the control group was similarly administered BSA antigen at the same dose. Booster immunization was administrated on day 21.

Geese were fed ad libitum with a mixed feed of 12.5% crude protein, supplemented with green grass whenever possible, as described previously [20]. Blood samples were randomly collected from 12 geese per group by venipuncture (wing vein) on days 1, 11, 21, 31, and 42. Plasma was obtained by centrifugation and stored at –20 °C until analysis of antibody titers. Antibody titers were measured by ELISA, as described previously [18].

### 2.4. Tissue Collection

On day 42 of the experiment, six individuals with hard-shell eggs in the uterus were slaughtered for tissue sample collection from each group. Ovarian follicles were classified as LWFs (white follicles, 4 mm < diameter < 8 mm), SYFs (yolky follicles, 8 mm ≤ diameter ≤ 10 mm), and LYFs (yolky follicles, diameter > 10 mm). As described previously [18], granulosa layers of ovarian follicles were separated from the five largest LYFs (F1: 48.5 ± 0.6 mm; F2: 42.4 ± 0.8 mm; F3: 33.7 ± 0.5 mm; F4: 23.8 ± 0.7 mm; F5: 14.4 ± 0.3 mm), SYFs, and LWFs. In addition, hypothalamus and pituitary gland tissues were collected. All tissue samples were snap-frozen in liquid nitrogen, and then stored at –80 °C until used.

### 2.5. RNA Isolation, cDNA Synthesis, and Quantitative Real-Time PCR

Total RNA was isolated using the RNAprep Pure Tissue Kit (Tiangen, Beijing, China) and then reverse-transcribed using the PrimeScript™ RT Master Mix (Takara, Kusatsu, Japan). The resulting cDNA was used for quantitative real-time PCR (qRT-PCR) using TB Green^TM^ Premix Ex Taq^TM^ II (Takara, Kusatsu, Japan). Some primers were obtained from our previous study, including *Gapdh*, *Gnrh1*, *Gnih*, *Fshb*, *Lhb*, *Lhr*, *Fshr*, *Star*, *Cyp11a1*, *Hsd3b*, *Ocln*, and *Smad4* [18]. Other primers were designed using Oligo 7 software, as shown in Table 1. Relative mRNA levels were calculated by the 2^−ΔΔCt^ method [21]. mRNA levels of target genes were normalized to the housekeeping gene *Gapdh*. Data were presented as mean ± standard error of mean (SEM).

### 2.6. Statistical Analysis

The data of antibody titers and mRNA levels of *Lhr*, *Star*, *Cyp11a1*, *Hsd3b*, *Ocln*, *Vldlr*, *Smad2*, *Smad3*, *Smad4*, and *Fshr* genes were transformed into logarithms before analysis because of great variation. Other data of mRNA levels of *Fst*, *Gnrh1*, *Gnih*, *Fshb*, and *Lhb* genes, follicle counts, and egg production were not transformed before analysis. Differences in mRNA levels of genes were analyzed using a two-way ANOVA with the classification of ovarian follicles and immunization treatment as the fixed factors. The two-way ANOVA was performed using Univariate GLM. For each group, differences in mRNA levels of each gene were analyzed using a one-way ANOVA followed by the Tukey’s multiple comparison test. An independent sample *t*-test was used for comparison between the two groups. A value of *p* < 0.05 was considered significant. All statistical analyses were performed using the IBM SPSS Statistics version 22 (IBM Corp., Armonk, NY, USA).

## 3. Results

### 3.1. Expression Pattern of Goose Fst Gene in the Granulosa Layer of Ovarian Follicles

The abundance of *Fst* mRNA declines as follicles mature in geese (Figure 1). *Fst* mRNA was the most abundant in SYFs, which was significantly higher than that in other follicles. The abundance of *Fst* mRNA in LWFs did not differ from amounts in F4 and F5 but differed from those in F1 to F3 (*p* < 0.05). No significant differences were observed among F1 to F5.

### 3.2. Expression of Recombinant Goose FST Protein

As an His-tag fused protein, recombinant goose FST protein was expressed with a predicted molecular mass of ~20 kDa. A main band of the recombinant goose FST protein was detected at the expected size of 20 kDa by SDS-PAGE (Figure 2A). The additional higher band with a molecular mass of 40 kDa was caused by dimer formation. Moreover, two immuno-reactive bands were observed by Western blot (Figure 2B).

### 3.3. Antibody Titer, Egg Production and Ovarian Follicular Development

After primary immunization, the anti-FST antibody titer level increased gradually in FST-immunized geese and was higher than that in BSA-immunized control geese on day 11 (*p* ˂ 0.05) (Figure 3A). The anti-FST antibody titer level continued to increase in FST-immunized geese after booster immunization and reached the peak on day 31 (Figure 3A). The egg-laying rate in FST-immunized geese was higher than that in BSA-immunized control geese after primary immunization (Figure 3B). During the entire experimental period, the cumulative number of eggs laid per goose was 10.6 for the FST-immunized group, whereas it was 7.7 for the BSA-immunized group (Figure 3C). On day 42 of the experiment, the ovarian follicles of both groups of geese were collected and classified (Figure 3D). Compared with those in BSA-immunized control geese, the number of LYFs was significantly higher in FST-immunized geese. No significant differences were observed in the number of SYFs and LWFs.

### 3.4. Gene Expression of Gnrh1, Gnih, Fshb, and Lhb in Hypothalamus and Pituitary Gland Tissues

Compared with those in BSA-immunized control geese, both *Gnrh1* and *Gnih* mRNA levels from the hypothalamus tissue were significantly higher in FST-immunized geese (Figure 4A,B). In the pituitary gland tissue, both *Fshb* and *Lhb* mRNA levels in FST-immunized geese were not significantly different from those in BSA-immunized control geese (Figure 4C,D). 

### 3.5. Gene Expression of Lhr, Star, Cyp11a1, Hsd3b, Ocln, and Vldlr in the Granulosa Layer of Ovarian Follicles

The abundance of *Lhr*, *Star*, *Cyp11a1*, and *Hsd3b* mRNA increased as follicles matured in both groups (Figure 5A–D). Compared with those in BSA-immunized control geese, *Lhr* mRNA levels in F5 and LWFs (*p* < 0.05, Figure 5A) and *Star* mRNA levels in F3, F5, SYFs, and LWFs (*p* < 0.05, Figure 5B) were higher in FST-immunized geese. Both the classification of ovarian follicles and immunization treatment influenced *Lhr* and *Star* mRNA levels. There was an interaction (*p* < 0.05) between these two variables for the *Star* mRNA level. Only the classification of ovarian follicles influenced *Cyp11a1* and *Hsd3b* mRNA levels (Figure 5C,D). 

The abundance of *Ocln* mRNA increased with follicular maturation but declined in F1 in both groups (Figure 5E). Only the classification of ovarian follicles influenced the *Ocln* mRNA abundance. The abundance of *Vldlr* mRNA decreased with follicular maturation in both groups (Figure 5F). Both the classification of ovarian follicles and immunization treatment had significant effects on *Vldlr* mRNA level; however, there was no interaction (*p* = 0.671) between these two variables. Compared with those in BSA-immunized control geese, *Vldlr* mRNA levels in SYFs and LWFs were significantly higher in FST-immunized geese (Figure 5F).

### 3.6. Gene Expression of Smad and Fshr in the Granulosa Layer of Ovarian Follicles

The abundance of the *Smad2*, *Smad3*, and *Smad4* mRNA increased as follicles matured in both groups, and then decreased in F1 (Figure 6A–C). Both the classification of ovarian follicles and immunization treatment influenced *Smad3* and *Smad4* mRNA levels; however, there was no interaction between these two variables. Only the classification of ovarian follicles influenced the *Smad2* mRNA level. Compared with those in BSA-immunized control geese, *Smad2* mRNA levels in F2 and F5 (*p* < 0.05, Figure 6A) and *Smad3* mRNA levels in F5, SYFs, and LWFs (*p* < 0.05, Figure 6B) were higher in FST-immunized geese. There were significant differences in *Smad4* mRNA levels in F2, F3, F4, F5, and LWFs between the two groups (Figure 6C). Compared with that in LWFs, the *Fshr* mRNA level increased in SYFs and then remained high in F1 to F5 in both groups (Figure 6D). Only the classification of ovarian follicles influenced the *Fshr* mRNA level. There was a significant difference in *Fshr* mRNA levels from F5 between the two groups.

## 4. Discussion

FST plays an important part in the regulation of ovarian follicular development by neutralizing effects of activin in mammals; however, only a few studies have explored the same in fowls. In this study, the abundance of *Fst* mRNA in ovarian follicles of Yangzhou geese were investigated first. This was found to be consistent with previous studies on other domestic fowls in which *Fst* mRNA abundance peaked in SYFs and then declined with ovarian follicular maturation. Following immuno-neutralization of FST bioactivity in geese, the number of LYFs significantly increased, indicating that the development potential of ovarian pre-hierarchical follicles into pre-ovulatory follicles had been enhanced. We further demonstrated that enhanced ovarian pre-hierarchical follicle development was associated with increased mRNA levels of genes involved in ovarian steroidogenesis and yolk deposition in FST-immunized geese.

Anti-FST antibodies gradually appeared in the blood circulation of geese as the titers increased after primary immunization and peaked following booster immunization. Corresponding to the appearance of anti-FST antibodies, both the egg-laying rate and the cumulative number of eggs laid per goose were higher in the FST-immunized group than those in the BSA-immunized group. It was noted that the egg-laying rates decreased in the BSA-immunized group at the early stage of the experiment, which might be due to the stresses caused by the experiment operation and antigen administration, as geese very easily suffered from stresses. However, the anti-FST antibody resisted the detrimental effect of such stresses and maintained egg production by prompting the maturation of LYFs in the FST-immunized group. An important observation was that there was an increase in the number of LYFs in FST-immunized geese. Considering that avian ovarian follicles develop in a size hierarchy, this result suggests that more LWFs were recruited and developed into LYFs through the SYFs. Pre-hierarchical follicles, including LWFs and SYFs, are gonadotropin-independent and mainly regulated by local ovarian growth factors, such as members of the TGF-β family [22]. Activin, a member of the TGF-β family, plays an important role in the promotion and maintenance of follicular growth during the early developmental stage [23]. In hen pre-hierarchical follicles, activin A induces SMAD signaling through its own membrane receptors type IIB and type I and stimulates the differentiation of granulosa cells [24,25]. Considering that FST inhibits activin bioactivity by forming an inactive complex with it, we hypothesized that immuno-neutralization of FST bioactivity enhances the developmental potential of LWFs and SYFs by increasing activin availability. This result was in line with previous findings in which the number of smaller follicles at the time of wave emergence increased in FST-immunized heifers resulting from the reduced neutralization of activin [15].

In hens, follicular activin A was mainly confined to the theca layer and was low in pre-hierarchical follicles; however, it increased sharply as follicles entered the pre-ovulatory hierarchy [26]. Similarly to activin A, FST predominated in the theca layer, although significant amounts were also present in the granulosa layer of pre-hierarchical follicles [26]. In vitro studies in hens have shown that activin A significantly increased mRNA levels of gonadotropin receptors *Fshr* and *Lhr* in granulosa cells of pre-ovulatory follicles [27]; however, it inhibited the proliferation of granulosa cells of pre-ovulatory follicles [27,28]. In this study, *Lhr* and *Fshr* mRNA levels were significantly upregulated in the granulosa layer of pre-ovulatory follicle F5 in FST-immunized geese, and the *Lhr* mRNA level was also upregulated in the granulosa layer of pre-hierarchical follicle LWFs. These results were similar to those of a previous study, in which activin A increased the expression level of *Lhr* mRNA, but not *Fshr* mRNA, in granulosa layer cells of pre-hierarchical follicle SYFs in hens [24]. These results suggest that FST plays a limiting role in the regulation of granulosa cell differentiation by neutralizing activin A, which promotes gonadotropin receptor expression.

For the pre-hierarchical follicles in the hen, the granulosa layer cells are undifferentiated and steroidogenically incompetent, and mRNA levels of *Lhr*, *Star*, *Cyp11a1*, and *Hsd3b*, the genes that are involved in progesterone synthesis, are low [29,30,31,32]. Once follicles enter the pre-ovulatory hierarchy, the *Star* mRNA level increases, and follicles begin to produce progesterone [29]. In this study, early granulosa cell differentiation and steroidogenic competency in pre-hierarchical follicles from FST-immunized geese could be seen from *Star* mRNA levels, which were found to increase significantly in SYFs and LWFs. A previous study in hens found that activin A directly induces *Lhr* mRNA expression and indirectly promotes *Star* mRNA expression and progesterone production following *Lhr* mRNA induction in SYFs [24]. Moreover, upregulated *Vldlr* mRNA levels were observed in SYFs and LWFs, indicating enhanced yolk deposition [33]. During the rapid growth phase, VLDLR migrates to the follicular wall, enabling endocytosis of yolk precursors into the oocyte, followed by follicular differentiation [34]. However, the abundance of *Vldlr* mRNA decreased with follicular maturation, which may allow the yolk precursor to reach the oocyte membrane directly by passing through intercellular gaps between granulosa cells for receptor-mediated endocytosis [35,36,37]. Until now, there has been no research on *Vldlr* gene regulated by activin A or FST. It was found that activin A induced *Ocln* mRNA expression in the granulosa layer of chicken ovarian follicles, resulting in functional tight junctions to prevent access of yolk into the oocyte [38]. However, immunization against FST had no effect on *Ocln* mRNA expression in this study. These changes in gene expression suggest that immuno-neutralization of FST bioactivity enhances the development potential of ovarian pre-hierarchical follicles, manifested as early granulosa cell differentiation, steroidogenic competency, and raised yolk deposition.

Activins perform their biological functions by forming heterodimeric complexes with type I and type II receptors, in which the type I receptor is trans-phosphorylated by the type II receptor [23]. The activated type I receptor then phosphorylates receptor (R)-SMAD2 and/or 3, which form complexes with Co-SMAD4. These activated SMAD complexes translocate to the nucleus to alter gene transcription. Immunization against FST significantly increased *Smad3* mRNA levels in SYFs and LWFs, which was consistent with the pattern of *Star* mRNA level. This result indicates that the effect of nullifying FST bioactivity to promote the growth and differentiation of pre-hierarchical follicles in geese is mediated by SMAD3 in the enhanced activin receptor signaling pathway. It is worth noting that the *Smad4* mRNA level was upregulated in both pre-hierarchical and pre-ovulatory follicles in FST-immunized geese. There is some evidence that FST can regulate ovarian function by forming an inactive complex with other members of the TGF-β superfamily, BMPs [6,39]. Studies in chicken ovaries have demonstrated that BMPs signaling via SMAD molecules promote granulosa cell differentiation and progesterone production [40,41,42]. Therefore, immunization against FST might enhance BMP signaling and result in an increase in progesterone synthesis through SMAD4, thereby accelerating the development of pre-ovulatory follicles in geese.

Unexpectedly, immunization against FST did not significantly increase the *Fshb* mRNA level from the pituitary gland, which is inconsistent with FST’s known involvement in the inhibition of FSH secretion in birds. Although plasma FSH concentration was not measured in this study, previous studies in heifers have found no change in plasma FSH concentration after FST immunization [15]. However, immunization against FST significantly upregulated *Gnrh1* and *Gnih* mRNA levels from the hypothalamus. In vitro studies had shown that activin A might act at the hypothalamus in mice and activate the secretion of GnRH by modulating *Kiss1* gene expression, whereas FST reduced *Kiss1* gene expression and abolished activin’s effect on the *Kiss1* gene [43]. More studies are needed to understand these findings in birds.

## 5. Conclusions

Immuno-neutralization of FST bioactivity enhanced the developmental potential of ovarian pre-hierarchical follicles by augmenting follicular sensitivity to activin, which resulted in increased numbers of pre-ovulatory follicles in Yangzhou geese. However, immuno-neutralization of FST bioactivity did not affect the pituitary *FSH beta* mRNA levels.

## Figures and Tables

**Figure 1 animals-12-02275-f001:**
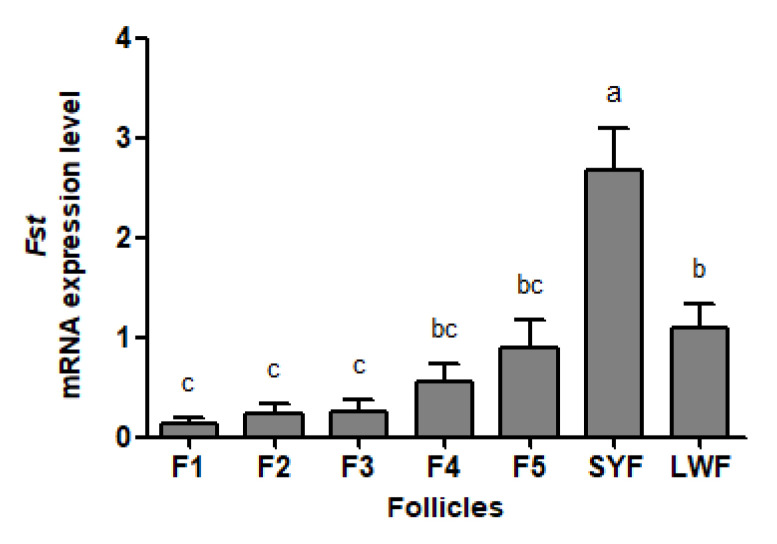
*Fst* mRNA abundance in the granulosa layers from ovarian pre-hierarchical follicles (SYFs and LWFs) and pre-ovulatory follicles (F1 to F5) in Yangzhou geese (*n* = 4). ^a,b,c^ represents a significant difference among the means (*p* < 0.05). Data represent mean ± SEM.

**Figure 2 animals-12-02275-f002:**
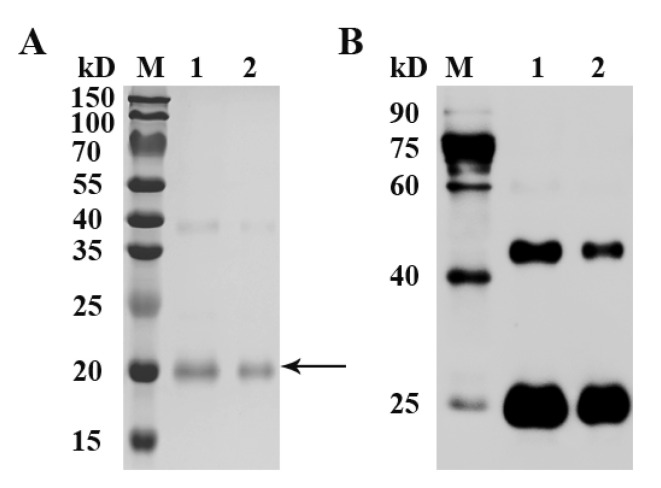
SDS-PAGE (**A**) and Western blot (**B**) analysis of recombinant goose FST protein. SDS-PAGE gel was stained with Coomassie blue. Arrow indicates the target protein. After incubation with an anti-His tag primary antibody followed by an HRP-conjugated secondary antibody, immuno-reactive bands were visualized with ECL reagent. M, protein standards; 1 and 2, purified proteins.

**Figure 3 animals-12-02275-f003:**
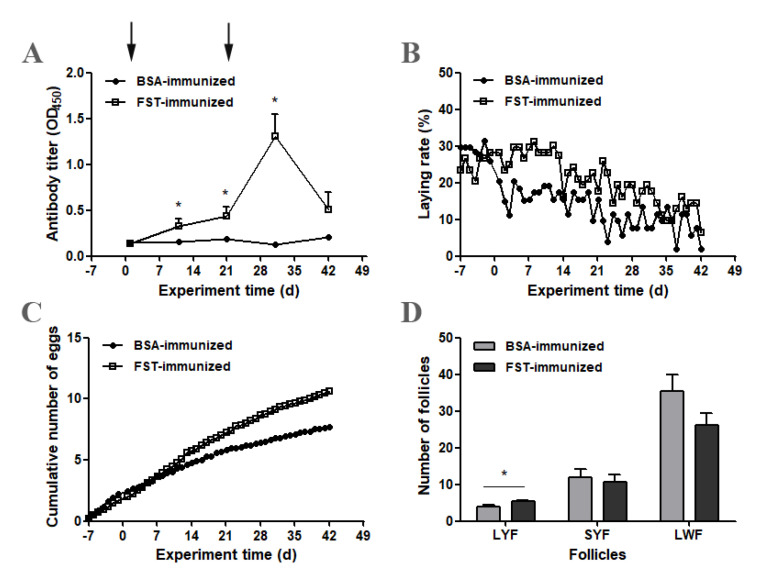
Changes in antibody titers, egg production and ovarian follicular development in FST-immunized or BSA-immunized control geese. (**A**) Anti-FST antibody titers (*n* = 12). (**B**) The laying rate represents the average laying rates over two consecutive days. (**C**) The cumulative number of eggs laid per goose equals the total number of eggs divided by the number of geese. (**D**) The number of ovarian follicles. * represents a significant difference between the two groups at one time point (*p* < 0.05). Data represent mean ± SEM.

**Figure 4 animals-12-02275-f004:**
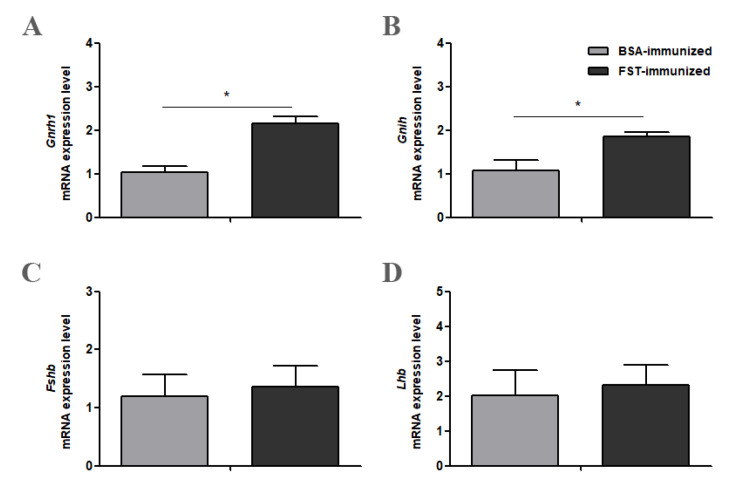
*Gnrh1* (**A**), *Gnih* (**B**), *Fshb* (**C**) and *Lhb* (**D**) mRNA levels in the hypothalamus or pituitary gland in FST-immunized or the BSA-immunized control geese (*n* = 6). * represents a significant difference between the two groups (*p* < 0.05). Data represent mean ± SEM.

**Figure 5 animals-12-02275-f005:**
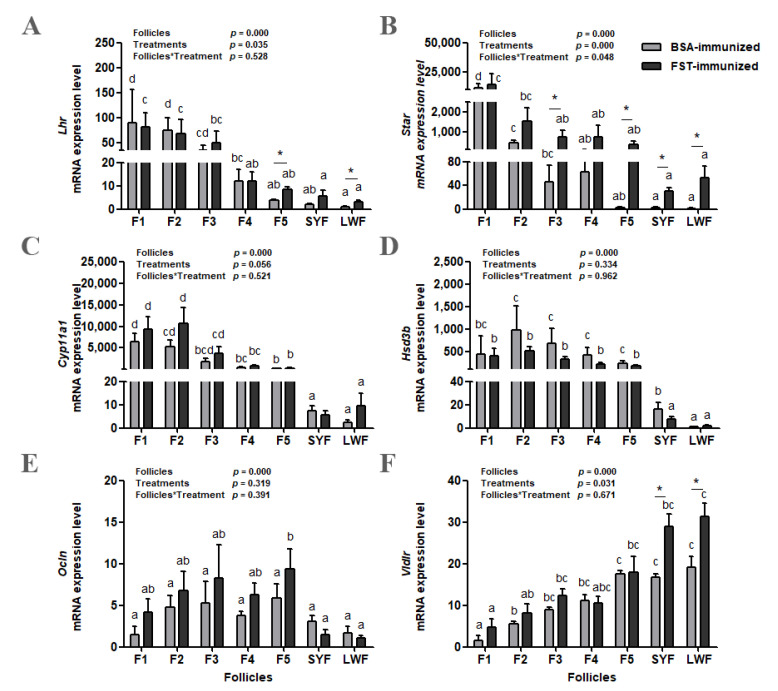
*Lhr* (**A**), *Star* (**B**), *Cyp11a1* (**C**), *Hsd3b* (**D**), *Ocln* (**E**), and *Vldlr* (**F**) mRNA levels in granulosa layers from ovarian pre-hierarchical follicles (SYFs and LWFs) and pre-ovulatory follicles (F1 to F5) in FST-immunized or BSA-immunized control geese (*n* = 6). For each group, ^a,b,c,d^ represents a significant difference among the means (*p* < 0.05). * represents a significant difference between the two groups in one type of follicle (*p* < 0.05). Data represent mean ± SEM.

**Figure 6 animals-12-02275-f006:**
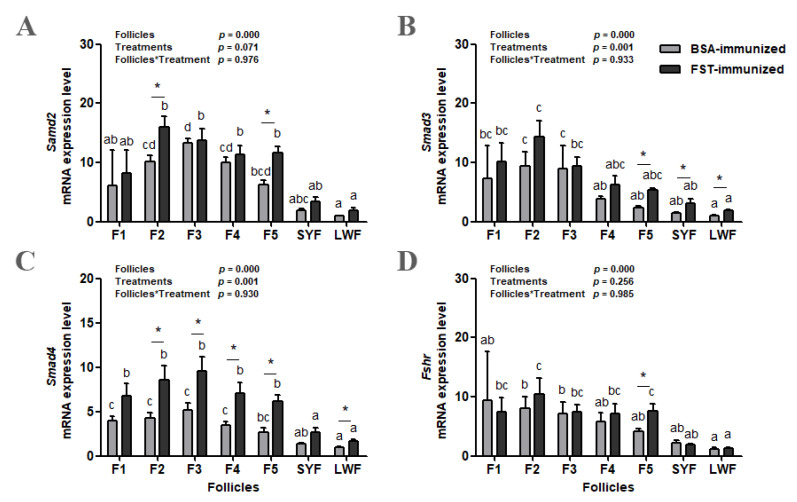
*Smad2* (**A**), *Smad3* (**B**), *Smad4* (**C**), and *Fshr* (**D**) mRNA levels in granulosa layers from ovarian pre-hierarchical follicles (SYFs and LWFs) and pre-ovulatory follicles (F1 to F5) in FST-immunized or BSA-immunized control geese (*n* = 6). For each group, ^a, b, c^ represents a significant difference among the means (*p* < 0.05). * represents a significant difference between the two groups (*p* < 0.05) in one type of follicle. Data represent mean ± SEM.

**Table 1 animals-12-02275-t001:** Primers used in the real-time PCR assay of genes.

Gene Name	Accession Number	Primer Sequences (5′-3′)	Annealing Temperature (°C)	PCR Product (bp)
*Fst*	XM_013180613.1	F: CAGCCCGAACTTGAAGTCCA R: TATGCCATCATTCCCGCAGA	60	180
*Vldlr*	XM_013198844.1	F: GGGGCTCATCACTCCAGTCCTTG R: GGCAGTGCAATGGTGTGAGAGAC	60	171
*Smad2*	XM_013183184.1	F: ACTTATTCAGAGCCTGCGTTC R: TGTAATAGAGGCGCACTCCC	60	223
*Smad3*	XM_013193229.1	F: CAGCCCTCTATGACCGTGGA R: AGGCACTCAGCAAACACCT	60	170

## Data Availability

Data are contained within the article.

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
