# Peer review of "Immuno-Neutralization of Follistatin Bioactivity Enhances the Developmental Potential of Ovarian Pre-hierarchical Follicles in Yangzhou Geese"

_animals, 2022, doi:10.3390/ani12172275_

Round 1
Reviewer 1 Report
The article entitled “Immuno-neutralization of Follistatin Bioactivity Enhances the Developmental Potential of Ovarian Pre-hierarchical Follicles in Yangzhou Geese” provides new findings of follistatin regulation of the goose ovarian follicular development. The authors showed that after immuno-neutralization of follistatin bioactivity, the number of ovarian preovulatory follicles increased, and mRNA levels of genes involved in ovarian steroidogenesis and yolk deposition were upregulated in the granulosa layer of pre-hierarchical follicles. The authors postulated that follistatin plays a limiting role in the development of ovarian pre-hierarchical follicles into preovulatory follicles.
The manuscript is well written. The hypothesis is well defined. Methods are appropriate, figures are informative. Results and discussion in a majority are explanative.
The reviewer addressed only few questions to the authors.
Line 99 why BSA was used as a control?
Line 251 is there perhaps any other reason for this result in BSA-immunized group?
Discussion Do authors attempt to discuss the results of follistatin neutralization of GnRH and GnIR expression?
Reviewer 2 Report
The manuscript described the effects of immune-neutralization of follistatin (FST) on the growth of ovarian follicles in Yangzhou geese. Results showed the inhibition of FST enhanced development of follicles in association with increase in progesterone synthesis and yolk precursor uptake related genes in granulosa layer cells. In addition, SMAD2, 3 and 4 genes, activin signaling pathway molecules, in granulosa layer of follicles were increased in FST neutralized group than control group. Authors concluded that the FST neutralization increases development of follicles by activation of activin in Yangzhou geese.
The manuscript shows detail of the function of FST on follicle growth in Yangzhou geese by inactivation of FST using immune-neutralization method. Therefore, the results are looks like in-direct evidence of effects of FST. In addition, the manuscript is well written, but the topic seems not general interest to audience. In addition, the mechanism is including some issues should be addressed, as below.
1. In this experiment, authors used immune-neutralization of FST for investigation of effects of FST. However, it seems difficult to prove whether the results obtained in this study are due to FST inactivation or FST stimulation. How do you think about the point?
2. In “Introduction” section, the purpose is lacked. Authors should add clear aim of the study. What is the purpose to investigate the effects of FST in the Yangzhou geese in this study?
3. Line 111-112. Authors should add the diameter of each F1 to F5 follicle.
4. Line 249-250. Egg laying rate is not increased in FST group in Figure 3B. Because, the laying rate in both groups is looks decreased compared with the start time point.
5. Authors should add the discussion about the result of OCDN.
6. Line 320-322. Authors discussed about synthesis of progesterone in this manuscript. How about authors analyze the progesterone concentration or indicator of increase in progesterone?
